# A Comparison of Oocyte Yield between Ultrasound-Guided and Laparoscopic Oocyte Retrieval in Rhesus Macaques

**DOI:** 10.3390/ani13193017

**Published:** 2023-09-26

**Authors:** Nadine Piekarski, Theodore R. Hobbs, Darla Jacob, Tiah Schwartz, Fernanda C. Burch, Emily C. Mishler, Jared V. Jensen, Sacha A. Krieg, Carol B. Hanna

**Affiliations:** 1Division of Reproductive and Developmental Sciences, Oregon National Primate Research Center, Oregon Health & Science University, Beaverton, OR 97006, USA; burchf@ohsu.edu (F.C.B.); mishlere@ohsu.edu (E.C.M.); jared.jensen@colostate.edu (J.V.J.); hannaca@ohsu.edu (C.B.H.); 2Animal Resources & Research Support, Oregon National Primate Research Center, Oregon Health & Science University, Beaverton, OR 97006, USA; jacobd@ohsu.edu (D.J.); schwatia@ohsu.edu (T.S.); 3Division of Reproductive Endocrinology and Infertility, Department of Obstetrics and Gynecology, Oregon Health & Science University, Portland, OR 97239, USA; kriegs@ohsu.edu

**Keywords:** non-human primate, oocyte retrieval, ultrasound-guided, laparoscopy, animal welfare, assisted reproductive technology

## Abstract

**Simple Summary:**

Research using Assisted Reproductive Technology (ART) in non-human primates (NHPs) is fundamental for improving human reproductive health as well as for generating models of disease. The first critical step for many studies utilizing ART is to obtain high-quality oocytes. Two techniques for oocyte retrieval are currently used at the Oregon National Primate Research Center (ONPRC): the standard surgical approach by laparoscopy and the less invasive ultrasound-guided approach. From a welfare perspective, ultrasound-guided oocyte retrieval provides many benefits; however, it has not been established whether this technique has a similar efficacy to laparoscopy. In our study, we compared an extensive data set on oocyte yield and fertilization rates from laparoscopic and ultrasound-guided oocyte retrievals in Rhesus macaques. Our analysis reveals that the ultrasound-guided technique is equivalent to the laparoscopic one in those aspects. In summary, the two techniques yielded statistically equivalent oocyte yields, therefore, the less invasive ultrasound-guided oocyte retrieval technique is recommended as a refinement that improves animal welfare.

**Abstract:**

Obtaining quality oocytes is a prerequisite for ART-based studies. Here we describe a method for transabdominal ultrasound-guided (US) oocyte retrieval in rhesus macaques (*Macaca mullata*) and compare it to the standard surgical approach using laparoscopy (LAP). We analyzed oocyte yield from six continuous reproductive seasons (2017–2023) that included *n* = 177 US-guided and *n* = 136 laparoscopic oocyte retrievals. While the ultrasound-guided technique retrieved significantly fewer oocytes on average (LAP: 40 ± 2 vs. US: 27 ± 1), there was no difference in the number of mature metaphase II oocytes (MII) between the two techniques (LAP: 17 ± 1 vs. US: 15 ± 1). We show that oocytes retrieved by the ultrasound-guided approach fertilize at the same rates as those obtained via the laparoscopic procedure (LAP Fert Rate: 84% ± 2% vs. US Fert Rate: 83% ± 2%). In conclusion, minimally invasive ultrasound-guided oocyte retrieval improves animal welfare while delivering equivalent numbers of mature oocytes, which are ideal for ART. Furthermore, we show that oocyte competency, as represented by fertilization rate, is not affected by retrieval technique. Therefore, the Oregon National Primate Research Center (ONPRC) has adopted the ultrasound-guided approach as the standard technique for oocyte retrieval.

## 1. Introduction

Nonhuman primates (NHPs), such as Rhesus macaques (*Macaca mulatta*), are important animal models for biomedical research because of their evolutionary closeness to humans [1]. Modeling human disease with NHPs has thus become a major focus of research, propelled forward by advancements in genome editing and ARTs [2,3]. Obtaining high-quality oocytes is the first crucial step in this endeavor.

Following controlled ovarian stimulation (COS), oocytes are recovered and assessed by their nuclear morphology to determine meiotic stages. These stages include: (1) immature oocytes arrested at prophase I of meiosis with a clearly visible germinal vesicle (GV); (2) immature oocytes that have undergone GV breakdown and have resumed meiosis to metaphase I (MI); and (3) mature metaphase II (MII) ova identified by an extruded polar body that are receptive to being fertilized (Figure 1).

In clinical settings, transvaginal ultrasound-guided follicle aspiration has been the gold standard procedure for recovering oocytes since its introduction in 1987 by Wikland [4,5]. Before ultrasound-guided retrievals were established, oocytes were routinely obtained by laparoscopy, sometimes with ultrasound assistance. As an alternative technique for oocyte retrieval in humans, ultrasound-guided percutaneous follicle aspiration has been employed; however, this is not as regularly used [5,6].

In Rhesus, oocyte retrieval after hormonal stimulation is commonly conducted by laparoscopy. The laparoscopic procedure in this species was first described in detail by Dierschke and Clark in 1976 [7] and has been proven very effective for obtaining quality oocytes for ARTs ever since [8,9,10]. While laparoscopy works well and is minimally invasive, it is still a surgical procedure that involves two or more skin incisions and penetration of the abdominal cavity. From a welfare perspective, ultrasound-guided oocyte retrieval is much less invasive and is not considered a surgical procedure [11]. Transvaginal ultrasound-guided oocyte retrieval is currently not possible due to anatomical differences between rhesus and humans. Compared to humans, the diameter of the vaginal vault is smaller in Rhesus, and the position of the ovaries is closer to the abdominal wall than to the cervix [12]. For these reasons, one promising alternative is percutaneous transabdominal ultrasound-guided oocyte retrieval, which has been previously performed successfully in Rhesus macaques [11,12,13] as well as in other NHPs such as baboons [14,15] and Squirrel Monkeys [16]. The ultrasound approach is less invasive because the animal does not require abdominal insufflation, incisions, or penetration, and there are no incisions to suture or to monitor for healing post-operatively. Follicles are well visualized using ultrasound, even those located beneath the ovarian cortex. While ultrasound technique and skill level are necessary as follicles are very small targets for the aspiration needle, the needle guide technology certainly simplifies the task. Furthermore, in regard to procedure time duration, the ultrasound-guided retrieval requires about half the time in comparison to the laparoscopic retrieval technique, which reduces anesthesia time and risk to the animal subjects.

It is currently unknown if ultrasound-guided follicular aspirations in Rhesus can be used to reliably recover oocytes of similar quantity and quality compared to the laparoscopic procedure. Specifically, we wanted to know if we could reliably retrieve mature ova, as in vivo matured oocytes have the highest developmental competence [17,18,19,20]. Therefore, the scope of this study was to determine if the ultrasound-guided retrieval technique could deliver similar numbers of MII ova with equal oocyte competency compared to the laparoscopic procedure. We describe our technique for optimized ultrasound-guided oocyte retrieval in detail and compare its efficacy with laparoscopic oocyte retrieval by empirically analyzing and comparing the oocyte yield and fertilization rates between the two.

## 2. Materials and Methods

### 2.1. Animals

A total of 111 regularly cycling female Rhesus macaques (*Macaca mulatta*) between 4 and 19 years of age underwent controlled ovarian stimulation and follicle aspiration, resulting in a total of 313 cycles. All animals were housed at the ONPRC, an American Association for Accreditation of Laboratory Animal Care (AAALAC) accredited institution, and cared for in line with the Guide for the Care and Use of Laboratory Animals. All animal procedures conducted at this facility were performed by trained veterinary and technical staff in accordance with the Public Health Services Policy on Humane Care and Use of Laboratory Animals and approved by the Institutional Animal Care and Use Committee (IACUC). A physical examination and medical records review were completed by a veterinarian for each macaque prior to this study. All macaques were deemed healthy with no evidence of preexisting conditions.

### 2.2. Controlled Ovarian Stimulation (COS)

Female rhesus macaques underwent COS cycles following a standardized protocol previously described [21,22]. Briefly, starting on days 1–4 of the menstrual cycle, females received 20–30 IU of recombinant human follicle-stimulating hormone (FSH) intramuscularly (IM) twice daily (BID) for 8 days (Organon, West Orange, NJ, USA). On days 7 and 8 of the COS cycle, a combination of the same dose of FSH with 30 IU luteinizing hormone (LH) was given IM BID (Serono Reproductive Biology Institute, Rockland, MA, USA). Serum concentrations of hormone levels for estradiol (E_2_) and progesterone (P_4_) were measured by the ONPRC Endocrine Technologies Core using a Roche cobas e411 (Roche Diagnostics, Indianapolis, IN, USA) to monitor ovarian follicle growth. Females were administered 0.75 mg/kg of the gonadotropin-releasing hormone (GnRH) antagonist, Acyline (NICHD Contraceptive Discovery and Development Branch, Bethesda, MD, USA), subcutaneously the following morning when reaching an E_2_ serum level concentration of 200 pg/mL to prevent an endogenous luteinizing hormone (LH) surge and ovulation prior to retrieval. To stimulate oocyte maturation and ovulation, females were administered 1100 IU of recombinant human chorionic gonadotropin IM (rhCG: Ovidrel, EMD Serono, Rockland, MA, USA). Females underwent follicle aspiration (FA) 36–38 h post-hCG administration if serum levels of E_2_ and P_4_ suggested an appropriate response to hCG on day 9 of the COS cycle (E_2_ ≥ 1000 and P_4_ ≥ 1.0).

### 2.3. Anesthesia and Patient Preparation

Animals were fasted overnight prior to surgery. Each subject was administered ketamine HCl (5 to 10 mg/kg IM) for sedation (Covetrus, Dublin, OH, USA). Anesthesia was maintained with isoflurane (Piramal Healthcare Limited, Boston, MA, USA) 1 to 2 Vol% in 100% oxygen administered through an endotracheal tube. Subjects were placed in dorsal recumbency, and one intravenous catheter (22-gauge intracath) was inserted in a cephalic vein. Lactated Ringer’s solution was administered intravenously throughout the procedures at a rate of 10 mL/kg/h. Animal temperature was maintained using a heated surgical table, a Gaymar water re-circulating heat pad, and a Bair Hugger forced air warming blanket. The abdomen of each subject was shaved to remove hair from xyphoid to the pubis, and ChloraPrep (2% *w*/*v* chlorhexidine gluconate and 70% *v*/*v* isopropyl alcohol, Becton Dickinson, El Paso, TX, USA) was applied in a circular pattern beginning in the center of the abdomen. Surgical personnel for each follicle aspiration procedure included one surgeon, one dedicated anesthetist, and one laboratory assistant for specimen handling and transport.

Analgesia following the procedures differed between the two techniques because of the less invasive nature of the ultrasound aspirations in comparison to the laparoscopy. For the laparoscopic method, animals received 0.2 mg/kg sustained release buprenorphine subcutaneously (ZooPharm LLC, Laramie, WY, USA) and 0.5 mL bupivacaine 0.5% with epinephrine (Hospira, Inc., Lake Forest, IL, USA) administered intradermally at the incision site. For the ultrasound-guided method, animals received 0.05 mg/kg buprenorphine (NW Compounders, Portland, OR, USA) administered IM prior to the procedure.

### 2.4. Laparoscopic Approach

After sterile skin preparation and surgical draping of the abdomen, a 5 mm skin incision was created approximately 1–2 cm cranial to the umbilicus. Subcutaneous tissue was bluntly divided to expose the midline rectus fascia. A Verres needle was inserted through the fascia and into the abdominal cavity. Through the Verres needle, the abdomen was then insufflated with CO_2_ to an intraabdominal pressure of 10–15 mm Hg. Each subject was maintained on a mechanical ventilator through the duration of abdominal insufflation. The Verres needle was removed, and a 5 mm-diameter trocar/cannula was advanced into the abdomen. After the removal of the trocar, a 5 mm rigid endoscope was advanced through the cannula, and the subject was placed in the Trendelenburg position. After initial visualization of the reproductive organs, a 5 mm diameter accessory port was placed caudolateral to the endoscope port at the level of the ovaries. Grasping forceps were advanced through the accessory port and used to manipulate and stabilize each ovary. For oocyte harvest, grasping forceps were used to secure either the uteroovarian ligament or the infundibulopelvic ligament. The ovaries were then turned and manipulated to allow visualization of the follicles. After stabilization of the ovary with the grasping forceps, the aspiration needle was inserted through the abdominal wall and into the follicles (Figure 2A,B). Each follicle was observed to collapse around the needle as the follicular fluid was aspirated via the vacuum system and contained within the collection vial. After follicular fluid was collected from all visible follicles on each ovary, the abdomen was lavaged and suctioned with warmed saline to remove any blood within the abdomen using an aspiration cannula inserted through the accessory port. Both ports were then removed, and each incision closed with a 4–0 monofilament absorbable suture, closing the rectus fascia with a simple interrupted suture followed by inverted simple interrupted sutures to close the skin incisions. Analgesia consisted of 0.2 mg/kg Buprenorphine Sustained-Release (ZooPharm LLC, Laramie, WY, USA) administered subcutaneously immediately prior to the procedure.

### 2.5. Ultrasound-Guided Approach

A GE Voluson E8 Expert ultrasound utilizing a 3D/4D convex probe with a needle guide was used for this study (Figure 2C). After sterile skin preparation, ultrasound gel was applied to the inner terminal end of a sterile probe cover. The probe cover was then extended to cover the probe and cable to allow maintenance of a sterile field. A total of 70% alcohol was then sprayed onto the abdomen, and the probe was used for a preliminary ultrasound examination of the reproductive organs. The machine settings were adjusted to maximize the contrast between follicular fluid and ovarian stroma. Follicles were seen as round or oval sonolucent structures. An estimated follicle count on each ovary was then obtained. Next, a sterile needle guide was positioned over the probe, which was integrated with the ultrasound software and provided a needle trajectory (Figure 2D, dotted line) on the viewing screen. After confirming the minimal distance between the abdominal wall and ovary and ensuring that no other structures would be injured by the needle, the entry point was determined. The probe was positioned over the ovary, and the aspiration needle advanced through the needle guide, along the trajectory line, and into the follicles (Figure 2D). Each follicle was visualized collapsing around the aspiration needle, and the needle was repositioned to the next follicle until all follicles were aspirated. Ideally, the skin and peritoneum were entered once to access the entire ovary to minimize trauma. This procedure was repeated on the contralateral ovary. At the conclusion of the procedure, ultrasound surveillance of the abdomen was used to confirm hemostasis of the ovaries bilaterally.

### 2.6. Follicle Aspiration (FA) and Sample Collection System

A central vacuum and vacuum regulator (Datex-Ohmeda, Madison, WI, USA) provided uniform suction of −100 mmHg for FAs. The aspiration system consisted of a 20 g hypodermic needle with PB hub (Cadence Science, Cranston, RI, USA), Teflon tubing (0.047″ I.D.), and a collection vial (15 mL centrifuge tube, Corning Inc., Corning, NY, USA) that contained TALP-HEPES supplemented with 3 mg/mL bovine serum albumin (BSA) and 10 USP/mL heparin to prevent blood clot formation [22]. The collection vials were maintained at 37 °C in a block heater (Fischer Scientific, Pittsburgh, PA, USA). Following FA, oocytes were transported in a portable incubator (Darwin Chambers, St. Louis, MO, USA) at 37 °C to the laboratory.

### 2.7. Oocyte Recovery and In Vitro Fertilization (IVF)

Once in the laboratory, aspirates were further diluted with TALP-HEPES supplemented with BSA and 3% hyaluronidase to remove cumulus cells [22]. Isolated oocytes were washed into fresh TALP-HEPES on a heated stage, and the remaining cumulus cells were removed by gentle pipetting to enable visualization of the oocyte nuclear configuration. Oocytes were sorted according to their nuclear stage of maturation (GV, MI, or MII, see Figure 1) and transferred into culture plates with equilibrated TALP and held until insemination.

Insemination was performed 6–7 h after oocyte retrieval by conventional in vitro fertilization (IVF), as described in Ramsey and Hanna 2019 [22]. Fertilization was checked 14–16 h post-insemination and confirmed when either 2 pronuclei and/or 2 polar bodies were observed (Figure 1D). The metaphase II fertilization rate was calculated using the number of fertilized MII from the day of retrieval. Embryos were moved into equilibrated culture dishes with BO-IVC culture medium (IVF Bioscience, Falmouth, UK) and kept in a humid incubator with a mixed gas supply that resembles physical conditions inside the body to improve outcomes (6% CO_2_, 5% O_2_, 89% N_2_) at 37 °C [23].

### 2.8. Statistical Analysis

The comparison of mean values between the two different groups of laparoscopic (LAP) and ultrasound-guided (US) retrievals was analyzed using an independent-samples *t*-test. The results are reported as mean ± standard error of the mean (SEM), unless stated otherwise. Any *p* value below 0.05 was considered statistically significant. All the results of the analysis are summarized in Table 1. Statistical analysis and graphs for figures were created using GraphPad Prism (San Diego, CA, USA, version 10.0.0).

## 3. Results

### 3.1. Comparison of Oocyte Yield and Quality Based on Retrieval Technique

We analyzed the oocyte yield over six reproductive seasons from 2017–2023, which included a transition from the laparoscopic approach to the ultrasound-guided technique in 2020. A total of *n* = 136 retrievals using laparoscopy and *n* = 177 ultrasound-guided oocyte retrievals were empirically evaluated. The results are summarized in Table 1 and Figure 3. The median age of both groups is comparable and is not expected to have an effect on the results represented in this study (Lap: 9.3 ± 0.2; US: 10.3 ± 0.2).

On average, significantly fewer oocytes were recovered using the ultrasound-guided approach in comparison to laparoscopy. Laparoscopic aspirations led to an average yield of 40 oocytes per retrieval, compared to 27 oocytes per retrieval for the ultrasound-guided technique (LAP: 40 ± 2 vs. US: 27 ± 1, *p* < 0.01). Upon further evaluation of the data, the decrease in overall oocyte number is manifested only by immature oocytes. The mean number of immature oocytes, both MI and GV, was significantly decreased with ultrasound-guided retrievals (MI LAP: 13 ± 1 vs. MI US: 6 ± 0, *p* < 0.01; GV LAP: 8 ± 1 vs. GV US: 4 ± 0, *p* < 0.01).

Importantly, the mean number of mature oocytes (MII) recovered did not change significantly between the two techniques (LAP: 17 ± 1 vs. US: 15 ± 1, *p* = 0.19). As a result, the percentage of mature oocytes (MII) to the total number of oocytes was significantly increased using the ultrasound-guided technique, while the percentage of immature oocytes (MI and GV) decreased significantly (MII LAP: 44% ± 2% vs. MII US: 52% ± 2%, *p* < 0.01; MI + GV LAP: 52% ± 2% vs. MI + GV US: 36% ± 2%, *p* < 0.01).

### 3.2. Fertilization

To assess the possible impact on oocyte quality, we compared the fertilization rates of the mature MII oocytes between the two retrieval techniques. We found no significant difference in the MII fertilization rates among the two groups (LAP: 84% ± 2% vs. US: 83% ± 2%, *p* = 0.64). However, we do see a significant difference in the mean numbers of fertilized oocytes between the two techniques. Due to in vitro maturation and the increased number of immature oocytes at the time of retrieval—in particular, MI stages—laparoscopic retrieval leads to a significantly higher number of fertilized oocytes in comparison to ultrasound-guided retrieval (LAP: 24 ± 1 vs. US: 15 ± 1, *p* < 0.01).

## 4. Discussion

NHPs are invaluable as non-human models of disease due to their anatomical and physiological similarities to humans. More specifically, ART research using NHP gametes is essential for studying reproductive health in humans. But when using animal models, researchers are obligated to find the least invasive methods in order to minimize the impact of their research on animal welfare. Here we analyzed the efficacy of the non-invasive transabdominal ultrasound-guided oocyte retrieval in rhesus macaques and compared it to the standard laparoscopic procedure.

Our study reveals that ultrasound-guided oocyte retrieval in rhesus macaques delivers a similar number of oocytes in the MII stage compared to the laparoscopic procedure. Mature oocytes (MII) at the point of retrieval are known to have the best developmental potential and are therefore the most valuable for IVF-based studies [17,18].

When the total number of oocytes at retrieval are compared, laparoscopy leads to an average of 32% more oocytes than ultrasound-guided it is worth noting that the number of oocytes retrieved impacts the quality of oocyte handling. Once follicles are aspirated, many time-sensitive handling steps such as isolating oocytes from aspirates, removing cumulus cells, and determining the stage of meiosis are conducted at the bench in atmospheric air before oocytes are moved into equilibrated culture dishes in the incubator. Oocytes are particularly vulnerable to temperature and pH fluctuations, which will affect spindle formation and development [24,25]. Therefore, a lower number of oocytes at retrieval will decrease the time needed to deal with them safely at the bench. In this respect, the ultrasound-guided technique is advantageous as it provides the same number of MII while delivering overall fewer oocytes.

Immature oocytes in the MI and GV stages make up for the difference in the total number of oocytes between the two retrieval techniques. Those immature oocytes are likely derived from aspirates of smaller follicles [26,27,28], which are better visualized using the laparoscopic procedure. A similar result has been found in humans comparing laparoscopic and ultrasound-guided oocyte retrieval between the same patients [29].

Our study demonstrates that both retrieval techniques lead to similar fertilization rates, providing evidence that oocyte competency is not affected by retrieval procedures. However, we do see a significant increase in the number of fertilized oocytes with the laparoscopic approach. This increase is due to the higher number of immature oocytes at the point of retrieval, especially those in the MI stage, which may undergo in vitro maturation (IVM) and become susceptible to fertilization. Although it appears more beneficial to use laparoscopy to obtain those additional fertilized oocytes, it is unclear if their developmental competence is similar to that of mature MII in vivo. It has been reported that invitro matured oocytes display a lower fertilization rate, which might be caused by cytoplasmic immaturity [17,18,20]. Furthermore, it is assumed that IVM and resulting MIIs may display artifacts in spindles and chromosomes that decrease their chance of developing into blastocysts [30,31,32]. Unfortunately, blastocysts from this study were integrated into other projects and could not be used to provide a better understanding of IVM and the resultant developmental competence.

## 5. Conclusions

The main goal of doing IVF is not necessarily to collect the highest number of oocytes possible but rather to obtain high-quality mature oocytes that have a good chance of fertilization and development into blastocysts. The ultrasound-guided technique is equivalent to the laparoscopic technique in this respect. The benefits of the ultrasound-guided technique include reduced anesthesia time, no insufflation of the abdomen, and no incisions, therefore no need for wound care, allowing for faster recovery and reduced analgesia needs. Because the number of surgical procedures for each individual animal subject is regulated, this refinement removes surgical procedures from protocols and preserves the animal resource. In addition to these animal welfare benefits, the increased proportion of mature oocytes delivered via the ultrasound-guided approach reduces laboratory time spent sorting oocytes and reduces the amount of disposable supplies required. Therefore, transabdominal ultrasound-guided oocyte retrieval has become the new standard for oocyte retrieval in Rhesus macaques at ONPRC and may potentially be applied to other species as well.

## Figures and Tables

**Figure 1 animals-13-03017-f001:**
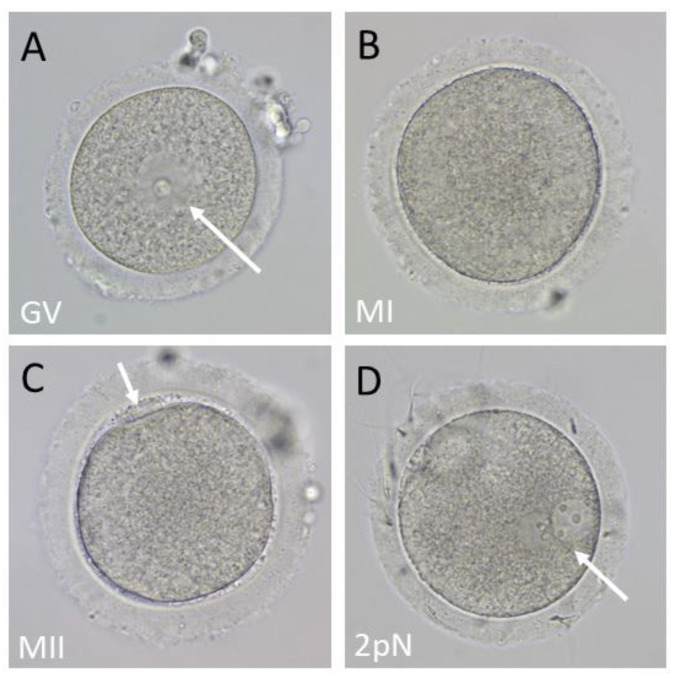
(**A**–**C**) Different stages of oocytes from Rhesus macaques were found at the time of retrieval in order of their stages of maturation. (**A**) An immature oocyte with a clearly visible germinal vesicle (arrow). (**B**) An immature oocyte that has resumed meiosis to metaphase 1 (MI). The germinal vesicle is broken down, and there are no other morphological markers. (**C**) The mature oocyte called MII is identified by an extruded polar body (arrow) and is susceptible to fertilization. (**D**) A fertilized oocyte (2pN) with two pronuclei indicated by an arrow.

**Figure 2 animals-13-03017-f002:**
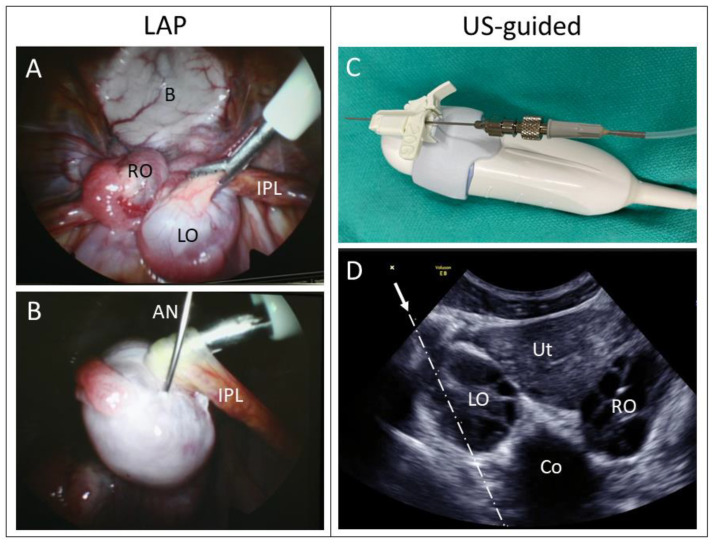
(**A**,**B**) Laparoscopic views into the Rhesus macaque pelvis:. (**A**) Forceps grasp the infundibulopelvic ligament (IPL) to stabilize the stimulated left ovary (LO). The right ovary (RO) and bladder (B) can be seen in the background. (**B**) The aspiration needle (AN) within a follicle of a stimulated ovary. (**C**,**D**) Pictures from an ultrasound-guided procedure. (**C**) The 3D/4D ultrasound transducer with an attached needle guide adaptor. The 20 ga. aspiration needle is inserted through the needle guide. (**D**) Ultrasound image of the uterus (Ut), right ovary (RO), left ovary (LO), and colon (Co). Note the honeycomb appearance of the multiple sonolucent follicles within the stimulated rhesus ovary. The arrow indicates the trajectory (dashed line) of the aspiration needle.

**Figure 3 animals-13-03017-f003:**
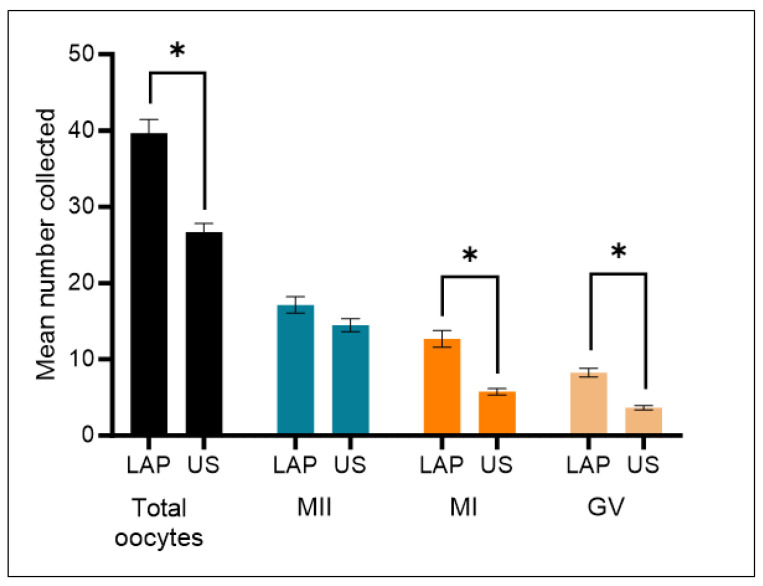
Comparison of oocyte yield from laparoscopic (LAP, *n* = 136) and ultrasound-guided (US, *n* = 177) retrievals. The mean number of total oocytes retrieved is significantly less using the US-guided approach. However, this result is only manifested by the numbers of immature oocytes (MI and GV) retrieved, while the mean number of mature MII oocytes is similar between both retrieval techniques. (* indicates a significant *p* value < 0.05).

**Table 1 animals-13-03017-t001:** A detailed summary of the results obtained from this study is presented as the mean ± standard error of the mean (SEM). A total of 313 cycles were analyzed, with 136 laparoscopic and 177 ultrasound-guided oocyte retrievals. Statistically significant values (*p* < 0.05) are indicated by an asterisk.

	Laparoscopic Retrieval	Ultrasound-Guided Retrieval	*p* Value
Total oocytes	39.7 ± 1.8 *	26.7 ± 1.1 *	<0.01
MII	17.1 ± 1.1	14.5 ± 0.8	0.19
MI	12.7 ± 1.1 *	5.7 ± 0.4 *	<0.01
GV	8.2 ± 0.6 *	3.7 ± 0.3 *	<0.01
% of MII to total # oocytes	44.3 ± 2 *	52.4 ± 1.9 *	<0.01
Number of fertilized oocytes	23.6 ± 1.4 *	15 ± 0.8 *	<0.01
MII Fertilization Rate (%)	84.3 ± 1.9	83.2 ± 1.5	0.64

## Data Availability

The data presented in this study are available on request from the corresponding authors.

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
