# Peer review of "A Comparison of Oocyte Yield between Ultrasound-Guided and Laparoscopic Oocyte Retrieval in Rhesus Macaques"

_animals, 2023, doi:10.3390/ani13193017_

Round 1
Reviewer 1 Report
The study aims of Piekarski et al. was to compare two techniques for oocyte retrieval in Rhesus macaques’ model: the standard surgical approach by laparoscopy (LAP) and the non-invasive ultrasound (US)-guided approach, concluding that the US-guided approach improves animal welfare while delivering equivalent numbers of MII oocytes that can be enrolled in ART procedures.
The results are coherently presented from a scientific point of view, accompanied by a careful study of the literature in the introduction section and motivated in the discussion and conclusion sections. I have only a few minor remarks and suggestions that I hope the authors will appreciate to improve an already satisfying work.
Minor Remarks
Abstract
Line 26: The acronym “ART” has been already used in the Simple Summary (Line 15) and in the Introduction section (Line 52). I suggest the authors to continue to use the established acronym throughout the main text.
Introduction
Line 60: The acronym “US” has been already used in the Abstract section (Line 27).
Materials and Methods
Line 114: I suggest the authors to control the way in which they need to insert citations (Stouffer and Zelinski-Wooten, 2004; Ramsey and Hanna 2019). In the main text, references must be numbered in order of appearance in the manuscript and the reference numbers should be placed in square brackets [ ], and placed before the punctuation (as the authors have already previously done, for example [1]; https://www.mdpi.com/journal/animals/instructions). I suggest the authors to maintain the same bibliographical approach throughout the main text.
Figure 1. A,B, C, D: It is only a stylistic suggestion; It would be nice to distinguish the two approaches (LAP and US-guided) by making two different panels. In my opinion, wrapping both procedures into one box can be a bit confusing.
Line 238: As previously specified, I suggest the authors to control the way in which they need to insert the references (Ramsey and Hanna (2019)). It is fine to state the author and date in the text if this contributes to greater clarity, however according to Animals “Author Guidelines” (https://www.mdpi.com/journal/animals/instructions), I think that the numerical reference should be inserted as well immediately after the “simple in-text citation”.
Results
Line 258: The authors wrote: “Table1”. Please provide a space between "Table" and "1".
Line 257: Check carefully all the acronyms throughout the manuscript. For e.g., the authors wrote the word "Laparoscopy" many times without using the mentioned acronym "LAP" (first mention in the abstract section, line 29).
Figure 2: It is only a stylistic suggestion; the authors could try to enlarge the figure, resizing the titles of the axes and the name of the experimental groups that, in my opinion, have been written using a font “too big”.
Table 1: The term “MII Fertilization rate (%)” is not entirely clear combined with the term “Number of fertilized oocytes” and this could lead to confusion. What do the authors mean by this term? The authors should better explain these results. I also suggest the authors to move the asterics (*) from the table headings to the “numerical values” on LAP and US-guided retrieval groups (For e.g. 39.7 ± 1.8*; 12.7 ± 1.1*; 52.4 ± 1.9* etc. etc.). Also, please check carefully all the acronyms throughout the manuscript (for e.g.: Ultrasound-guided retrieval ïƒ US-guided retrieval, etc etc).
Discussion and conclusions
Please check carefully all the acronyms.
Reviewer 2 Report
This manuscript described a method for transabdominal ultrasound-guided (US) oocyte retrieval in rhesus macaques and compare it to the standard surgical approach using laparoscopy (LAP). The results is reliable and very helpful to understand the meaning included. However, we in the developing country are very care and pay attention to the costing. Is it possible to exlain or analysis the costing? The cost of instruments, auxiliary equipments, and even solution are please included. For clinic use, we are also very care about the ease of operation and protocol simplicity.
Reviewer 3 Report
This manuscript systematically compared the number and quality of the oocytes retrieved by two different methods: 1) ultrasound-guided and 2) laparoscopic. The results will be informative to the field but more details on the analyses would need to be provided. Specific comments are as below:
1. it is stated that " female Rhesus macaques between 4 100 and 19 years of age" were used for this study. This is significant age span and it will be good that the authors clarify for each data what ages of the females were used and how many, and whether any significant differences were found between different age groups.
2. page 6, why did the authors use hypoxia condition? some references can be cited for this incubation method.
3. it makes sense that the ultrasound guided method will yield higher percentage of matured oocytes than laparoscopic. This may also indicate that the ultrasound guided method allows the female monkeys to be used for more times on experiments. Have the authors compared this aspect?
other specific comments:
1. staging criteria of the oocytes should be provided, the authors should add exemplified images on different stages of the oocytes (MII, MI, GV).
2. the "immature oocytes" can be misleading to differentiate oocytes before GV stage and oocytes before MII stage. The authors should specify more on what stages of the oocytes they got.
